# Thermodynamic evidence for a dual transport mechanism in a POT peptide transporter

**Joanne L Parker[1], Joseph A Mindell[2], Simon Newstead[1]***

[1]Department of Biochemistry, University of Oxford, Oxford, United Kingdom; [2]Membrane Transport Biophysics Unit, National Institute of Neurological Disorders and Stroke, National Institutes of Health, Bethesda, United States

**Abstract** Peptide transport plays an important role in cellular homeostasis as a key route for nitrogen acquisition in mammalian cells. PepT1 and PepT2, the mammalian proton coupled peptide transporters (POTs), function to assimilate and retain diet-derived peptides and play important roles in drug pharmacokinetics. A key characteristic of the POT family is the mechanism of peptide selectivity, with members able to recognise and transport >8000 different peptides. In this study, we present thermodynamic evidence that in the bacterial POT family transporter PepT$_{St}$, from *Streptococcus thermophilus*, at least two alternative transport mechanisms operate to move peptides into the cell. Whilst tri-peptides are transported with a proton:peptide stoichiometry of 3:1, di-peptides are co-transported with either 4 or 5 protons. This is the first thermodynamic study of proton:peptide stoichiometry in the POT family and reveals that secondary active transporters can evolve different coupling mechanisms to accommodate and transport chemically and physically diverse ligands across the membrane.

***For correspondence:** simon. newstead@bioch.ox.ac.uk

**Competing interests:** The authors declare that no competing interests exist.

## Introduction

Secondary active transporters are integral membrane proteins that couple the energy stored in an ion gradient to drive the uptake of a solute against its concentration gradient (*Nicholls and Ferguson, 2013*). This can be accomplished through either a symport mechanism, with the solute being moved in the direction of the driving ion, or antiport, where the solute movement is counter to that of the driving ion (*Shi, 2013*). The ion gradients utilised by secondary active transporters include proton ($\Delta\mu H^+$), sodium ($\Delta\mu Na^+$), or chloride ($\Delta\mu Cl^-$) gradients, which are in turn established through the action of the primary ATP driven P-, F-, V-, and A-type ion pumps (*Voet and Voet, 2011*). A fundamental characteristic of these systems is that, in general, transport is strictly coupled; the movement of solutes and ions is obligatory and one cannot be transported without the other. If these systems were to operate in a decoupled manner, they would act as leaks and dissipate the ion gradients across the membrane, quickly leading to cell death. Given the strict requirement for coupling solute binding and transport to ion movement, the stoichiometry of these mechanisms is normally a fixed ratio. Examples include the *Escherichia coli* lactose transporter, LacY, which transports lactose in a symport mechanism with one proton (*Kaback et al., 2011*) and EmrE, the small multidrug extrusion transporter, which moves both monovalent and divalent substrates in a 1:2 drug:proton stoichiometry (*Rotem and Schuldiner, 2004*).

A number of membrane proteins have been identified that recognise multiple structurally and chemically diverse solutes (*Koepsell, 2013*; *Pelis and Wright, 2014*). Prominent among these are the proton coupled oligopeptide transporters or POTs (*Hillgren et al., 2013*; *Smith et al., 2013*). POT family transporters are widely distributed within bacterial, fungal, and plant genomes where they are responsible for the uptake of di- and tri-peptides from the external environment (*Daniel et al., 2006*).

**eLife digest** The cell membrane encases cells and functions as a protective barrier. Although this has the benefit of preventing harmful substances from entering a cell, it also keeps beneficial molecules out. The cell membrane therefore contains a system of different 'gates', called transporters, through which selected supplies can pass.

One large family of transporters, found in bacteria, mammals, and plants, is the 'proton coupled oligopeptide transporter' family, called POTs for short. These transport over 8000 types of small peptide molecule, each of which is made up of two or three smaller molecules called amino acids. The energy for this transport process is gained by simultaneously transporting charged ions called protons with the peptides. Because these transporters also recognize and transport various drugs, they are currently being investigated to discover whether they could be manipulated to increase how much of a drug is taken up into cells.

It remains unknown how the POT family of transporters imports so many different small peptides across the cell membrane, or how many protons are needed to transport a peptide. A study published earlier in 2014 has nevertheless provided some hints: it appears that small peptides adopt different shapes when bound to a bacterial POT transporter depending on whether they consist of two or three amino acids. This suggests that two different transport mechanisms operate from the same binding site, which may account for the wide variety of molecules that can be transported.

In a follow up to this work, Parker et al., including some of the researchers involved in the earlier 2014 work, now look in detail at how many protons this bacterial transporter uses to import these small peptides. This reveals that while the transport of peptides made of three amino acids requires three protons to also be moved through the transporter, the transport of peptides containing two amino acids requires four, or possibly five, protons. This challenges previous findings that these transporters transport one peptide for every proton, and further supports the idea that a single transporter can use more than one method to bind to and transport molecules. Whether other membrane transporters, particularly the human versions of the POT family, share this ability remains an open question.

Mammals contain four POT family transporters, PepT1 (SLC15A1), PepT2 (SLC15A2), PHT1 (SLC15A4), and PHT2 (SLC15A3). PepT1 and PepT2 are expressed at the plasma membrane, whereas PHT1 and PHT2 are found in lysosomal membranes (*Daniel and Kottra, 2004*). Throughout the POT family the transport mechanism and peptide binding site are highly conserved, with bacterial counterparts sharing ~80% identity to human PepT1 and PepT2 within their peptide binding sites (*Terada and Inui, 2012*; *Newstead, 2014*). All POT family members studied to date transport their substrates into the cell in a coupled symport mechanism, driven by the proton electrochemical gradient. While a number of mutational studies on the mammalian PepT1 and PepT2 transporters address peptide recognition (*Terada et al., 1996*; *Fei et al., 1997*, *1998*; *Yeung et al., 1998*; *Uchiyama et al., 2003*; *Luckner and Brandsch, 2005*; *Kulkarni et al., 2007*; *Pieri et al., 2009*), the question of how many protons are coupled to peptide transport remains unresolved; early studies using Caco-2 cell lines derives a ratio of greater than two protons per peptide (*Thwaites et al., 1993*). However due to experimental design, narrowing this figure to a more precise stoichiometry was not possible (*Kottra et al., 2002*). Electrophysiological studies using two electrode voltage clamping (TEVC) in Xenopus oocytes in tandem with radio ligand transport assays on non hydrolysable peptide (D-Phe-L-Gln/Glu/Lys or Gly-Sar) have reported stoichiometry ratios of 1:1 and 2:1 proton:peptide for neutral/basic and acidic di-peptides respectively for PepT1 (*Fei et al., 1994*; *Steel et al., 1997*; *Chen et al., 1999*). Similar experiments on PepT2 have given different ratios either D-Phe-L-ala of 2:1 and for D-Phe-L-Glu 3:1 (Chen JBC 1999) or 1:1 for D-Phe-L-Gln/Glu or Lys (*Fei et al., 1999*).

Recently, we reported two crystal structures of a bacterial POT family transporter, PepT$_{St}$, from *Streptoccocus thermophilus*, which revealed di- and tri-peptides interacting differently within the binding site (*Lyons et al., 2014*). Whereas the di-peptide L-Ala-L-Phe binds in a horizontal position with respect to the plane of the membrane, whilst the tri-peptide L-Ala-L-Ala-L-Ala resides in a vertical orientation and makes subtly different interactions within the binding site. This raised the interesting and to our knowledge unique proposition, that two different transport mechanisms may have evolved

within the same binding site as a way to accommodate a diverse library of peptide ligands, >8000 (*Ito et al., 2013*). To address whether PepT$_{St}$ could indeed operate using distinct mechanisms to drive di- and tri-peptides, we explored the coupling mechanism between protons and peptide in a reconstituted system, determining the coupling stoichiometries of protons and peptides. We show that whilst tri-peptide import is coupled to three protons, the mechanism for di-peptide import requires at least four and possibly five protons. These results provide further biochemical evidence that POT family transporters do operate via multiple mechanisms for coupling peptide transport to the proton gradient. The ability to couple different numbers of protons to structurally and chemically diverse ligands may explain how the POT family is able to accommodate and transport such a large library of di- and tri-peptides.

## Results

To address the question of stoichiometry, we developed a sensitive and robust assay to follow the proton movement during the transport cycle. Previously published peptide transport assays tend to follow peptide uptake using a radiolabeled peptide substrate. To instead follow proton movement, we monitored the internal pH with the ratiometric pH sensitive fluorophore, pyranine (*Figure 1A* and *Figure 1—figure supplement 1*). We first performed a number of control experiments to see whether our system could indeed follow proton coupled peptide transport into a liposome. Acidification of the lumen was only observed in the presence of peptide and a large hyperpolarized (negative inside) membrane potential imposed by adding the potassium ionophore valinomycin in the presence of a K$^+$ gradient (*Figure 1B,C* and *Figure 1—figure supplement 2*). We did not see such acidification either in the absence of valinomycin or in the presence of amino acid (alanine) or tetra peptide (Ala-Ala-Ala-Ala), confirming that this transporter is indeed specific for di- and tri-peptide substrates (*Figure 1C*). PepT$_{St}$ has been shown previously to transport Ala–Ala with a 10-fold higher activity than Ala-Ala-Ala as judged by IC$_{50}$ values using tritiated Ala–Ala as a reporter (*Solcan et al., 2012*). We confirmed this 10-fold difference in uptake by using our proton-based assay, which can now report the direct uptake of any transported substrate rather than inferring substrate specificity through inhibition of Ala–Ala (*Figure 1—figure supplement 3*). This assay is also useful to study poor competitors of di-alanine for example, di-lysine where no competition could be observed previously (*Solcan et al., 2012*). Using this assay, we can now observe uptake of this substrate indirectly by monitoring the coupled proton movement (*Figure 1C*).

Since PepT$_{St}$ is an electrogenic transporter, we predict that imposition of a membrane potential in the presence of a pH gradient should drive uphill substrate transport. By loading the liposomes with high concentration of peptide and imposing a large hyperpolarising membrane potential (negative inside), we observe acidification of the lumen, indicating that the voltage can drive PepT$_{St}$-mediated transport against a 100-fold peptide gradient (*Figure 2A*). Importantly, we can also manipulate this system to see protons leaving the liposomal lumen as would be expected under an oppositely orientated membrane potential (positive inside). With an assay system set up where we could drive transport in predicted directions, we were now in a position to assess proton:peptide stoichiometry by measuring the equilibrium potential for proton flux using pyranine at a series of membrane voltages, set at the start of the assay with the appropriate potassium ion concentration gradient and the addition of valinomycin. This type of assay was used previously to address the stoichiometry of a lysosomal Cl−/H+ antiporter, CLC-7 (*Graves et al., 2008*).

We assume that PepT$_{St}$ operates via the coupled mechanism, nH$_{out}$ + mPep$_{out}$ ⇔ nH$_{in}$ + mPep$_{in}$, where the relative stoichiometry of protons:peptide is n/m. For this coupled system, the equilibrium potential is defined as the voltage at which there is no net substrate flux. This voltage, also known as the reversal potential (ΔΨ), is independent of the reaction mechanism. Rather, it depends only on the concentrations of protons and substrate and on the coupling stoichiometry with the form: ΔΨ = 60{[pH$_{in}$ − pH$_{out}$] − m/n log ([Pep]$_{in}$/[Pep]$_{out}$)}, where m and n are the stoichiometric coefficients in the chemical reaction above and ΔΨ is in mV (see derivation in *Figure 2—figure supplement 1*). For a given combination of pH and peptide gradients this equation predicts a voltage at which no net pH change will occur; voltages above and below that value should produce inward or outward proton flux, depending on the voltage. Conversely, if at a series of voltages (set using K$^+$/valinomycin), we observe acidification/no flux/alkalinzation, we can derive the relative stoichiometry of PepT$_{St}$ for protons and peptide. We performed such experiments for the neutral peptide, Ala-Ala-Ala with no net pH difference between the inside and outside of the liposome and a 100-fold peptide gradient

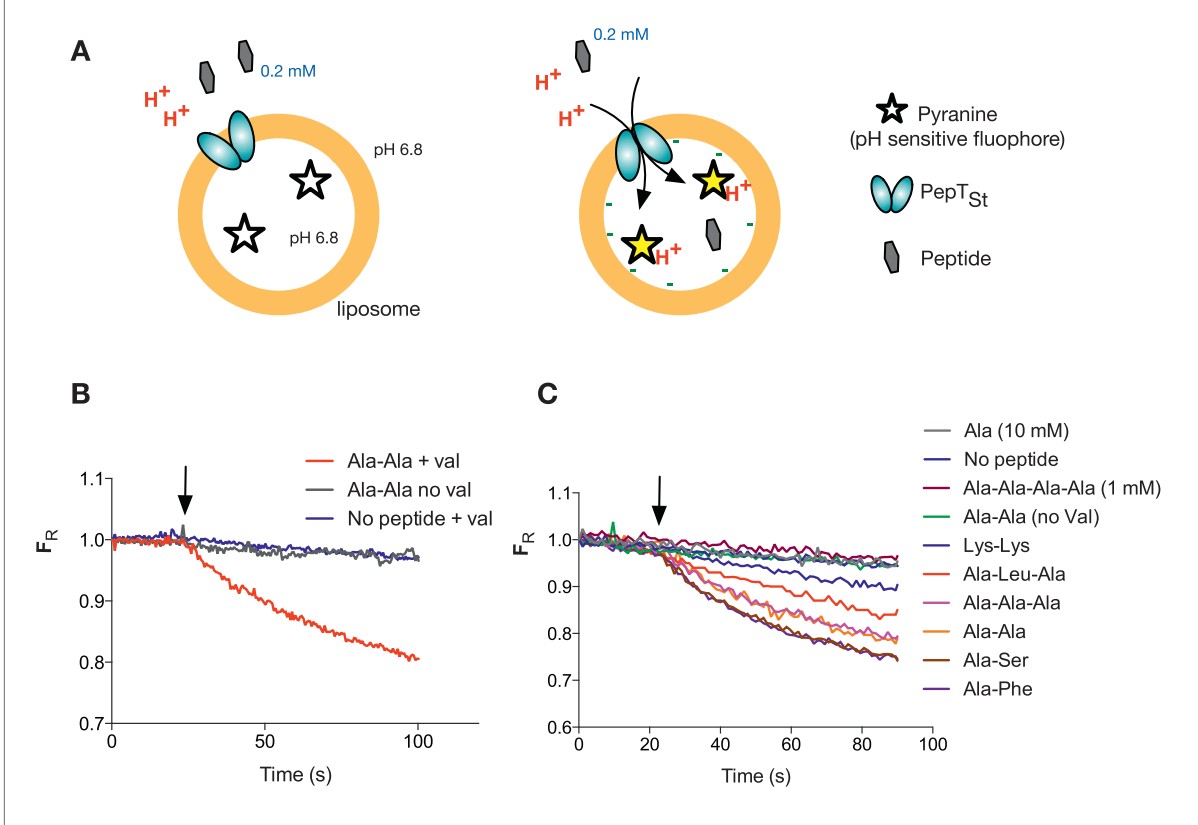

**Figure 1**. Monitoring peptide-coupled proton transport using the pH sensitive dye, pyranine. (**A**) Experimental setup to monitor proton flux. PepT_St is reconstituted into liposomes loaded with pyranine and a high concentration of potassium ions (120 mM), the external solution contains peptide and a low potassium concentration (1.2 mM). On addition of valinomycin, the membrane becomes highly potassium permeable, generating a hyperpolarised membrane potential (negative inside) this drives the uptake of peptide with protons, protonating the pyranine dye and altering its fluorescent properties. (**B**) Representative pyranine fluorescence traces produced from the set up described in (**A**) indicating that acidification of the liposomal lumen only occurs in the presence of a valinomycin (black arrow)-induced membrane potential with peptide. The Y axis indicates the fluorescence ratio as stated in the methods. (**C**) PepT_St can only transport di- and tri-peptides. To initiate transport, valinomycin was added to all experiments at the time indicated by the black arrow and the external substrate concentration was 0.2 mM. Data were normalised to the first time point for ease of comparison.

The following figure supplements are available for figure 1:

**Figure supplement 1**. Representative raw data of the pyranine fluorescence traces.

**Figure supplement 2**. PepT_St POPE:POPG proteoliposomes can hold a pH gradient of 1 unit.

**Figure supplement 3**. Transport strength of Ala–Ala vs Ala-Ala-Ala.

(higher concentration inside) and observed an absence of proton flux at a membrane potential of −40 mV (inside negative) which corresponds to a 3:1 proton:peptide stoichiometry (**Figure 3A** and **Figure 3—source data 1**). In contrast, voltages corresponding to reversal potentials for stoichiometries of 2:1 and 4:1 produced clearly distinguishable inward and outward fluxes respectively, strongly pointing to a 3:1 stoichiometry for the transporter. A very different combination of proton and peptide gradients, where we now included a proton gradient, also (pH more acidic outside) gave the same stoichiometry of 3:1 for the same tri-alanine peptide (**Figure 3B**). We also obtained this three proton:peptide stoichiometry for a different tri-peptide substrate, Ala-Leu-Ala (**Figure 3C**).

We went on to determine the proton:peptide stoichiometry of PepT_St when transporting di-peptide substrates. However, when we performed experiments with the same gradients as our initial tri-peptide measurements only now with the neutral peptide Ala–Ala, we still observed transport at a voltage of −40 mV, where before we saw no net proton flux for tri-alanine. Transport is still also

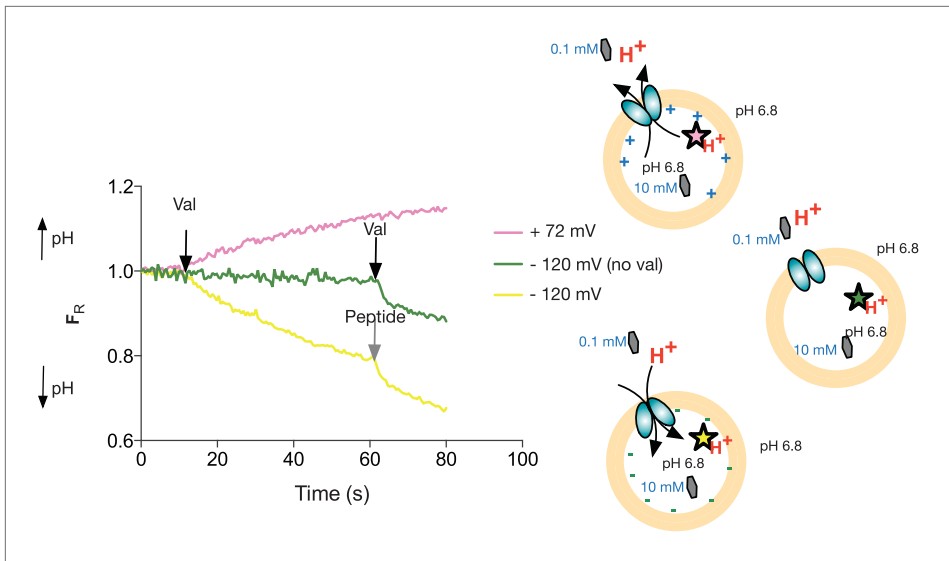

**Figure 2**. Peptide transport depends on the imposed voltage. Proteoliposomes in the presence of a 100-fold peptide gradient (0.1 mM outside and 10 mM inside) and no pH gradient (pH 6.8 both inside and outside). No proton flux occurs until the transmembrane potential is shunted by addition of valinomycin (green trace, val added at black arrow). $PepT_{St}$ can drive transport of peptides (and protons) against this gradient into the interior of the liposome using the proton electrochemical gradient when a negative voltage is imposed by valinomycin addition. (negative inside—yellow line). Further addition of peptide (0.5 mM) results in additional uptake (grey arrow). When a large (inside) positive voltage is applied (same peptide and pH gradients), peptides (and protons) exit the liposome (pink line).

The following figure supplement is available for figure 2:

**Figure supplement 1**. Derivation of the transport equation for a proton peptide transporter.

occurring at −30 mV which under these conditions would correspond to a proton:peptide stoichiometry of 4:1 and where we saw proton influx for tri-alanine (***Figure 4A***). Therefore, surprisingly, the stoichiometry of protons to peptides in $PepT_{St}$ appears to be different for tri-Ala as compared with di-Ala. Further experiments to try to pin down the number of protons being co-transported with di-alanine lead to slightly ambiguous results, as increasing proton:peptide stoichiometries predict decreasing increments in reversal potential. These are, in turn, harder to generate reproducibly with our valinomycin/$K^+$ system. We performed experiments with voltages set at reversal potentials predicted for symport ratios of 5 and 6 protons:peptide and proton flux was minimal at −20 mV (6 protons) but it is hard to be able to fully distinguish. Importantly, the system can still be driven in the opposite direction with higher voltages (0 mV, ***Figure 4B***). Regardless of whether the actual stoichiometry is 5 or 6 protons:peptide, these experiments strongly suggest that the stoichiometry is higher for di-peptides than for tri-peptides. We confirmed this higher ratio for di-peptide transport using the substrate Ala–Phe (***Figure 4C***). Again transport is clearly observed at a membrane potential at −30 mV, so the stoichiometry for di-peptide transport by $PepT_{St}$ is greater than four protons, clearly different from that of tri-peptides.

The reversal potential equation above predicts that if $\Delta\Psi = 0$, then $[Pep_{in}]/[Pep_{out}] = ([H^+_{out}]/[H^+_{in}])^n$, where n is the number of protons transported per peptide. Therefore, if our results are truly indicative of higher coupling ratios for di-peptides than for tri-peptides, the same proton electrochemical gradient should accumulate di-peptides to a higher steady-state concentration than tri-peptides. We tested this prediction by measuring the uptake of radiolabeled di- and tri-peptides in the presence of a fixed, 1-unit pH gradient (acid outside). As shown in ***Figure 5***, we find dramatically higher uptake of the di-peptide in this gradient, conclusively supporting the idea that different length peptides couple to the proton gradient with differing stoichiometries.

A different coupling stoichiometry for di- vs tri-peptides raises an interesting question of whether different amino-acid side chains within the transporter are required for di-peptide vs tri-peptide

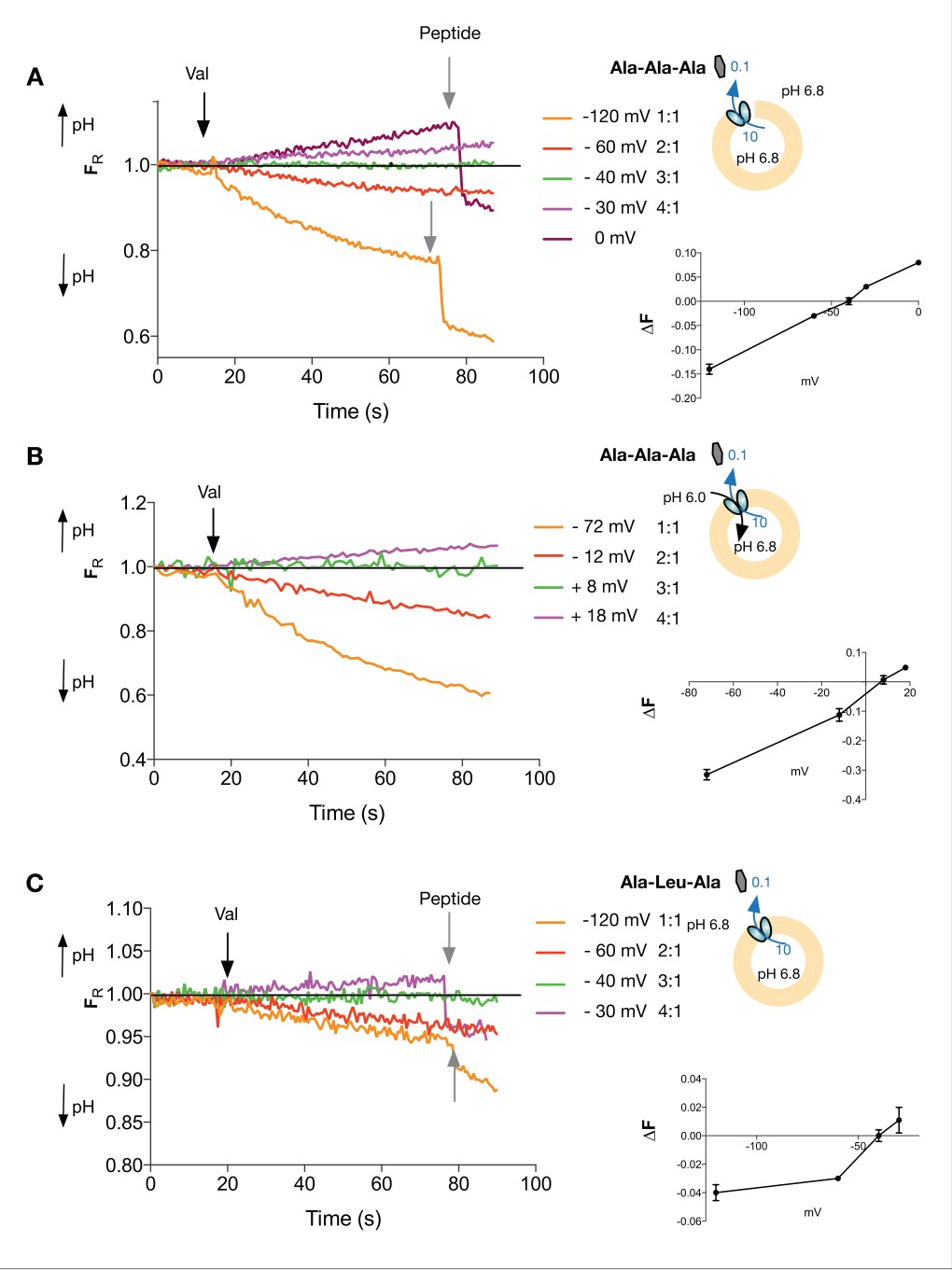

**Figure 3**. Tri-peptides are co-transported with three protons. The potassium gradient across the liposomes was varied in order to set the desired voltages (on valinomycin addition, black arrow) to achieve no net proton movement (indicated by the black line at $F_R$ of 1.0). The number next to the voltages are the proton:peptide stoichiometry that would reverse at that voltage. The internal peptide concentration (10 mM) (Tri-ala for **A** and **B**, and Ala-Leu-Ala for **C**) was 100-fold above that of the external concentration and the pH was inside 6.8, outside 6.8 (for **A** and **C**) and 6.0 (for **B**). Representative traces are shown for each experiment, which were repeated at least three independent times. The line graph for each experiment represents the mean change in fluorescence at time point 60 s and S.E.M. is indicated. Grey arrows indicate the addition of more peptide, to show that the system can be further manipulated.

The following source data is available for figure 3:

**Source data 1**. Table showing the conditions used to calculate the stoichiometry of transport.

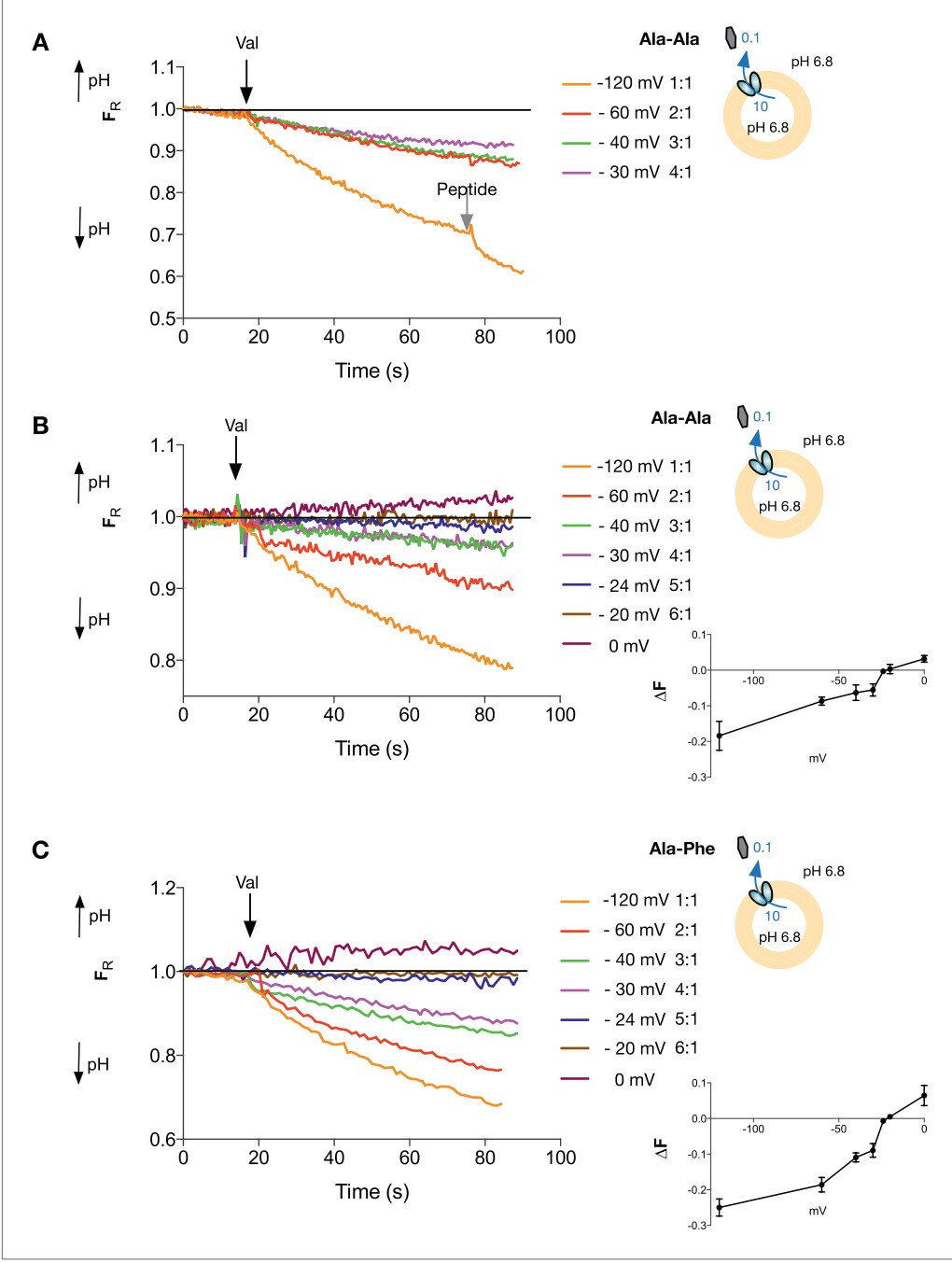

**Figure 4**. Di-peptide transport requires more protons than tri-peptide. The potassium gradient across the liposomes was varied in order to set the desired voltages (on addition of valinomycin, black arrow) to achieve no net proton movement (indicated by the black line at $F_R$ of 1.0). Only voltages that indicate a stoichiometry of proton:peptide of greater than 5:1 showed either no net movement of protons or reversal for both Ala–Ala (**A**, **B**) and Ala–Phe (**C**) di-peptides. Representative traces are shown for each experiment, which was repeated at least three independent times. The line graph for each experiment represents the mean change in fluorescence at time point 60 s and S.E.M. is indicated. Grey arrows indicate the addition of more peptide, to show that the system can be further manipulated.

transport. $PepT_{St}$ contains six-protonatable side chains within its binding site (Glu 22, 25, 299, 300, 400, and K126, **Figure 6A**). All of these with the exception of Glu299 are conserved across the PTR family from bacteria through to mammalian PepT1 and PepT2. Previous biochemical studies have

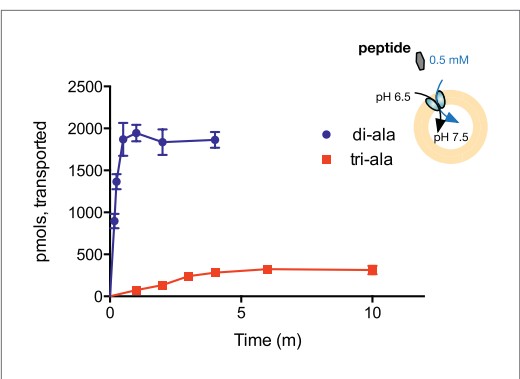

**Figure 5**. Steady-state accumulation of di- vs tri-alanine. Peptide transport was driven by an inwardly directed proton gradient in saturating amounts of peptide. Uptake was measured via scintillation counting using radiolabeled peptides ($^3$H for di-alanine and $^{14}$C for tri-alanine) and converted to pmols peptide transported per unit time.

shown that in PepT$_{St}$ this non-conserved residue is likely to be involved in structural and/or stability features specific to this protein as mutation of this residue results in no expression of the protein. Glu400 and Lys126 are likely to form a salt bridge that stabilises the outward open confirmation of the transporter, a feature that would be conserved regardless of the substrate. Glu300 has been shown to interact with both a di-peptide and a tri-peptide substrate and therefore likely to be involved in the transport mechanism for both di- and tri-peptides and has been shown to be important for di-alanine transport previously (*Solcan et al., 2012*; *Doki et al., 2013*). This leaves Glu22 and Glu25 as candidates for differential effects on di- and tri-peptide transport. Previously these residues have been shown to be important for proton coupling for di-alanine, however, here we also found that mutating either of these residues to alanine yielded proteins unable to catalyse proton coupled tri-alanine transport (*Figure 6B*). Therefore, despite different proton:peptide stoichiometries are apparently required for di- and tri-alanine transport, all five protonatable side chains within the binding site of PepT$_{St}$ are likely to be important for the transport mechanism.

## Discussion

### A dual transport model for proton coupled peptide symport in the POT family

A fundamental aspect of any transport mechanism is its coupling stoichiometry, how many ions are moved for each molecule of solute, as this information is necessary to generate reasonable mechanistic models for the transporter under study. Previous electrophysiological recordings on PepT1 and PepT2 have focused their attention on di-peptide substrates and suggest a 1:1 proton:peptide stoichiometry in PepT1 for neutral peptides and either 1:1 or > in PepT2 (*Smith et al., 2013*). Recent crystal structures and functional data on a bacterial POT family transporter, PepT$_{St}$, a homologue of PepT1 and PepT2, revealed that peptides could adopt different orientations within the binding site (*Figure 6C*). Whereas tri-alanine was observed adopting a vertical position and coordinated by 4 hydrogen bonds, the di-peptide L-Ala-L-Phe was held in a more horizontal position and coordinated through a more extensive network of interactions involving electrostatic interactions with conserved side chains from the N- and C-terminal bundles (*Lyons et al., 2014*). This raised the possibility that PepT$_{St}$ could transport peptides using two different mechanisms operating within the same binding site and has important implications for understanding proton coupled transport more generally within the POT family.

Here, we used a reconstituted proteoliposome system to accurately measure reversal potentials for peptide-coupled proton fluxes and found that di- and tri-peptides are transported using different proton stoichiometries. Assuming that our measurements on two sets of distinct di- and tri-peptides reflect the stoichiometries in general, our new data add to our previous multiple binding mode model by showing that tri-peptides are transported using three protons, whereas di-peptides are transported using four or possibly even five protons per cycle. It is important at this stage to highlight that our experiments cannot discriminate between protons that come through the transporter and those that may come through bound to the peptide. However, even if some protons are being moved on the peptide as opposed to being required to rearrange interaction networks during transport, our results still demonstrate that different numbers of protons are moved during the coupled transport of neutral di- vs tri-peptides, which we ascertain establishes a fundamental difference in the way this protein handles these two ligands.

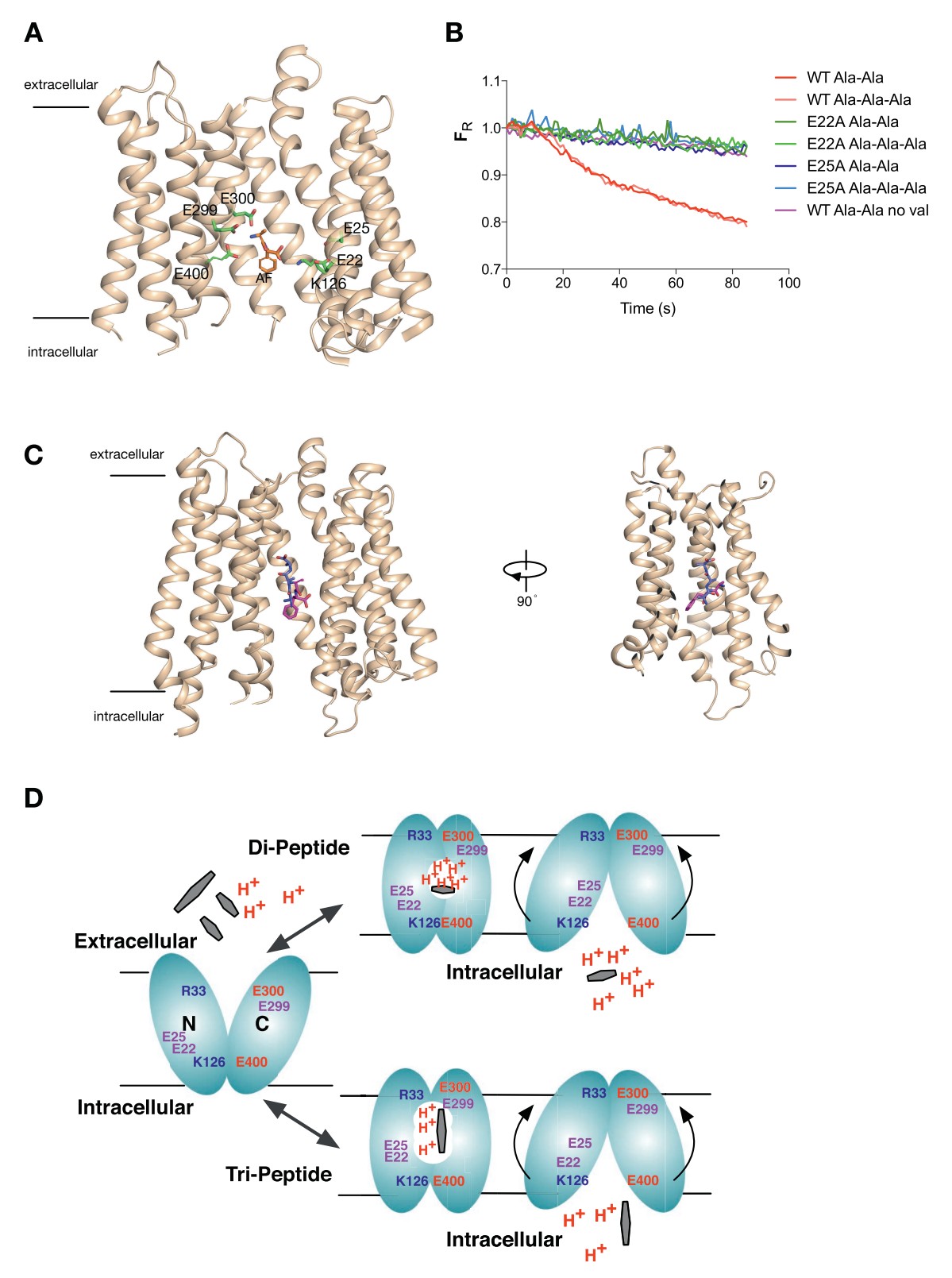

**Figure 6**. Model for proton:peptide symport. (**A**) Crystal structure of PepT$_{St}$ with bound Ala–Phe peptide (orange) (PBD 4D2C) showing the protonatable side chains within the binding site (green). Helices TM5 and TM8 have been removed for clarity. (**B**) E22A and E25A variants of PepT$_{St}$ are unable to couple proton movement to the transport either di-alanine or tri-alanine. (**C**) Ala–Phe and Ala-Ala-Ala adopt different orientations within the binding site
*Figure 6. Continued on next page*

*Figure 6. Continued*

of PepT$_{St}$. Two views of PepT$_{St}$ shown in the plane of the membrane and rotated 90°, with Ala-Ala-Ala (blue) and Ala–Phe (magenta) shown as sticks. Helices TM5 and TM8 have been removed for clarity. (**D**) Model for proton:peptide symport in PepT$_{St}$. Di-peptides transport requires at least four protons, whereas tri-peptides require only three, suggesting this is the lowest number of protons required to drive the conformational changes required for alternating access transport. Essential residues are indicated; residues involved in the potential stabilising salt bridges are labelled in blue and red, whereas protonatable side chains are labelled in purple.

The following figure supplement is available for figure 6:

**Figure supplement 1**. Model of the outward facing state of PepT$_{St}$ with bound Ala-Ala-Ala.

Interestingly, all five of the protonatable residues within the binding site are required for transport as none could be mutated and still allow for the transport of either peptide. This could be due to movement of the three protons within the binding site to different side chains, perhaps to help re-orientate the tri-peptide as the transporter transitions through its conformational cycle. Indeed if you overlay the crystal structure of the tri-alanine structure with a model of the outward open structure (*Figure 6—figure supplement 1*), tri-alanine in this position would obstruct the closing of the intracellular gate through disruption of the intracellular stabilising salt bridge (formed between Lys126 (TM4) and Glu400 (TM10)). Therefore, we suggest that for tri-alanine to be transported across the membrane, it is likely to undergo a change in its vertical binding position, to allow for closure of the transporter.

The question then arises as to the functional role of the protons in the transporter. We have previously identified six-protonatable side chains present in the binding site of PepT$_{St}$ (*Figure 6A*), with all but Glu299 being conserved across the POT family (*Solcan et al., 2012*). It would be tempting therefore, on the basis of simplicity, to predict that our data suggest only three side chains are protonated during tri-peptide transport, and four/five in di-peptide transport. However, we do not believe this to be the case. Our attempts to systematically remove the protonable side chains Glu22 and Glu25 on TM1, which form part of the conserved ExxERF motif but do not interact with either of the di-peptide or tri-peptide in the crystal structures (*Lyons et al., 2014*), resulted in inactive transporters (*Figure 6B*). Previous studies have shown that mutating any of the remaining side chains; Lys126, Glu300, and Glu400 also result in inactive proton coupled transport (*Solcan et al., 2012*). We conclude from these results that all of the protonatable side chains are required for transport regardless of substrate. However, the observation that tri-peptides can be moved using only three protons delineates the minimal number of de-protonation events that are required to drive the conformational changes that re-orientate the binding site. In this model, the remaining protonatable groups remain proton-bound throughout the transport cycle when tri-peptides are transported. On the basis of the new data presented here, we can add further mechanistic insight into our earlier model for peptide transport that we have summarised in *Figure 6D*. Alternating access within the POT family is physiologically driven by the proton electrochemical gradient, with defined conformational states stabilised through conserved pairs of salt bridges that act to coordinate the opening and closing of the intracellular and extracellular gates (*Newstead, 2014*). Starting from the outward open conformation the binding site is accessible to the extracellular side of the membrane, and the intracellular gate is closed and stabilised by a possible salt bridge between Lys126 (TM4) and Glu400 (TM10). Functional studies have revealed that in another bacterial POT family transporter, from *Geobacillus kaustophilus*, GkPOT, that the equivalent glutamate to Glu300 is protonated and may be required to allow the binding of peptide (*Doki et al., 2013*). Considering the minimal three-proton model for tri-peptide transport we propose, it seems reasonable to suggest that proton transfer from Glu300 to Glu400 during transport may occur to couple closing of the extracellular gate with opening of the intracellular one. This would account for one proton. The other two we suggest may come from Lys126 and either of Glu22 or Glu25. Our evidence is that in previous functional studies we showed that either of these side chains could be mutated to alanine with only a slight reduction in counterflow transport but complete loss of proton coupled peptide uptake (*Solcan et al., 2012*), behaviour classically used to identify side chains required for proton coupled uptake. In the case of di-peptides, additional deprotonation is clearly required. We conjecture that this is the result of the tighter coordination observed in the di-peptide complex structure compared to that for the tri-peptide (*Figure 6C*) and maybe one reason why this binding site is so sensitive to mutation in our assays.

An adaptable coupling mechanism, such as we propose, might have been an important component that enabled the POT family to adapt its binding site to accommodate structurally and chemically diverse molecules for nutritional assimilation. Whilst in bacterial, fungi, and mammals POT family homologues are responsible for peptide uptake, in plants this family has evolved to recognise widely diverse molecules, including nitrate, glucosinylates, hormones, and peptides (*Léran et al., 2014*; *Sun et al., 2014*; *Parker and Newstead, 2014*). This may explain why mammals use the POT family homologues PepT1 and PepT2, as coupling peptide transport to the proton gradient appears to facilitate a promiscuous binding site that can adapt to chemically diverse side chain groups more easily than sodium coupled transporters, which require well-defined binding sites for the cation.

## Materials and methods

### Reconstitution of PepT$_{St}$

For reconstitution, PepT$_{St}$ purified in the detergent DM (*Solcan et al., 2012*) was mixed in a 60:1 ratio (lipid:protein) with lipid vesicles composed of a mixture of POPE and POPG (in a 3:1 ratio). These lipids were chosen as they had been previously reported to form proton tight liposomes (*Tsai and Miller, 2013*). We confirmed this in our liposomes (*Figure 1—figure supplement 1*). The protein:lipid mix was diluted into a large volume of reconstitution buffer (50 mM potassium phosphate 6.8), and proteoliposomes were harvested by ultracentrifugation (>200,000×$g$ for 3 hr. Pelleted liposomes were resuspended at 0.5 µg/µl (protein) and dialysed extensively against reconstitution buffer (24 hr with two changes of buffer). Proteoliposomes were recovered and subjected to three rounds of freeze thawing before storage at −80°C.

### Transport assays

Proteoliposomes were harvested and resuspended in inside transport buffer (5 mM Hepes pH 6.8, 2 mM MgSO$_4$, 1 mM Pyranine (trisodium 8-hydroxypyrene-1,3,6-trisulfonate) also containing the desired potassium concentration (KCl) and peptide concentration) and subjected to three rounds of freeze thaw in liquid nitrogen and then extruded through a 0.4-µm membrane. Pyranine is a fluorescent pH indicator dye which is water soluble and can be trapped within liposomes. Acidification of the lumen of the liposome is indicated by a decrease in the ratio of fluorescence measured at 510 when excited at either 460 or 415. After extrusion the liposomes were harvested and excess pyranine removed through gel filtration using a superdex-25 column pre-equilibrated in inside transport buffer without pyranine. For the assays the liposomes were diluted into external transport buffer in a 0.85-ml micro cuvette with a small magnetic flea (5 mM HEPES pH 6.8 or 5 mM MES pH 6.0, 2 mM MgSO$_4$ and the desired amount of KCl to obtain the desired potassium gradient, ionic strength was kept equal across the liposome using NaCl). Transport was initiated using 1 µM valinomycin, and fluorescence was read at excitation 460 and 415 emission 510 in a Cary eclipse fluorimeter with continual stirring. To examine the data, the data were exported into Graphpad and the fluorescence was measured at 510 excitation 460 divided by that measured at 415 excitation, indicated at $F_R$ in the results. To compare multiple conditions, the data were normalised to 1 (from the first reading) for each experiment. Representative raw data are shown in *Figure 1—figure supplement 2*. For each individual experiment, the mean value was calculated from 55 to 65 s and this was repeated for each replicate (minimum of three) to generate an overall mean and S.E.M, which is plotted as a line graph on each figure.

### Transport assays using radiolabelled peptide

Proteoliposomes were harvested and resuspended in inside buffer (5 mM HEPES pH 7.5, 2 mM MgSO$_4$, 75 mM KCl) and subjected to three rounds of freeze thaw in liquid nitrogen and then extruded through a 0.4-µm membrane. For the assays, the liposomes were diluted into external transport buffer (5 mM MES pH 6.5, 2 mM MgSO$_4$ 75 mM KCl). Peptide, to a final concentration of 0.5 mM containing a tracer amount of either $^3$H-di-alanine (specific activity 30 Ci/mmol) or $^{14}$C-tri-alanine (specific activity 55 mCi/mmol), was added with 1 µM valinomycin and time points taken. The assays were performed at 22°C. Time points were taken and stopped by addition into 2 ml 0.1 M LiCl and filtering immediately through a 0.4-µm membrane in a vacuum manifold. The filters were washed twice with 2 ml of LiCl prior to scintillation counting in Ultima Gold (Perkin elmer). The amount of peptide transported into the liposomes was calculated based on specific activity for each peptide as detailed by the manufacturer and counting efficiency for the radioisotope in Ultima Gold counted in a Wallac scintillation counter ($^3$H 45% counting efficiency, $^{14}$C 98%). Experiments were performed four times to generate an overall mean and S.E.M.

## Acknowledgements

SN is a Wellcome Trust Investigator (102890/Z/13/Z). JAM is funded through the NINDS Intramural Program. We thank Chris Mulligan for helpful comments on the manuscript.

## Additional information

### Funding

| Funder | Grant reference number | Author |
|---|---|---|
| Wellcome Trust | 102890/Z/13/Z | Simon Newstead |

The funder had no role in study design, data collection and interpretation, or the decision to submit the work for publication.

### Author contributions

JLP, Conception and design, Acquisition of data, Analysis and interpretation of data, Drafting or revising the article; JAM, SN, Conception and design, Analysis and interpretation of data, Drafting or revising the article

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
