## [Decision Letter]

Thank you for sending your work entitled “One transporter, two mechanisms: Thermodynamic evidence for a dual transport mechanism in a POT peptide transporter” for consideration at *eLife*. Your article has been evaluated by John Kuriyan (Senior editor) and 2 reviewers. The editor and the reviewers find that the paper is of interest, and that it may be suitable for publication in *eLife*. They do, however, have some concerns about whether the principal conclusion, that the transport of dipeptides and tripeptides is coupled to different stoichiometries of protons, is based on a completely reliable interpretation of the data. Please consider the point referred to in the review below as major concern No. 1, and respond to us by email to the eLife editorial office telling us how you might deal with this particular issue. The editor and the reviewers will consider your response before reaching a decision on the paper.

*Review*:

The aim of the study presented by Newstead and colleagues is to determine the stoichiometry of a bacterial peptide/H+ symporter. The authors used a ratiometric fluorescence dye to record pH changes in liposomes. They demonstrated that the transporter transports protons when there is a voltage difference across the membrane, mediated by K+ and valinomycin. They also found that proton transport depends on the presence of peptides (Figure 1). The stoichiometry between peptides and protons was determined by finding the voltage at which H+ flux is zero (defined as the reversal potential) at known peptide and proton gradients. By relating the chemical potential difference of the peptide inside and out to the electrochemical driving force of the protons (which will depend on the number of protons per peptide), the stoichiometry (m/n) can be determined from the equation described.

This manuscript adds mechanistic novelty to Newstead's recent structures of proton-coupled peptide transporter family (POT) proteins by establishing an intriguing oddity concerning the transport mechanism of the two types of peptide substrates handled by PepT-type transporters. The previous crystal structures of the *S. thermophilus* PepT had shown that two classes of substrates, neutral di-peptides and tri-peptides, bind in different ways to the transport region. This raised questions about how the transporter, which handles a huge diversity of di- and tripeptides, manage this feat of nonspecificity within and specificity among substrate classes. Now, using an elegant assay in a reconstituted liposome-flux system, the authors show an important difference in H+-peptide stoichiometry between dipeptides (i.e., Ala-Ala) and tripeptides (Ala-Ala-Ala or Ala-Leu-Ala). This implies, under the assumption of strict H+ coupling for both substrate types that substantially separate transport mechanisms have evolved within the same protein. Given that assumption, that is a novel insight.

The authors also present experiments to identify specific residues that may be involved in handling the “extra” protons for the dipeptides. They mutate various protonatable residues that are seen to be close in the structures to the dipeptides, in hopes of finding mutants that abolish H+ coupling for the dipeptides, while preserving it for the tripeptides. Although no such specific residues emerged from these mutants, the effort and its “negative result” is nevertheless worthwhile to report.

*Major concerns*:

1) As written, there remains a major logical soft-spot in the central conclusion. The manuscript is properly explicit in stating the basic assumption underlying the stoichiometry measurement: that the transporter is fully, obligatorily coupled for both substrates. It is this assumption that leads to the equation on p 5 for the reversal potential, the zero-flux condition at thermodynamic equilibrium. If this assumption is valid, then all the conclusions offered here follow as the night the day. But what if it isn't? An alternative possibility is that the “less active” dipeptide substrate might show a higher H+/peptide stoichiometry because it is imperfectly coupled, and that 2 protons “slip through” on average for every transport cycle in which 3 protons are actually mechanistically coupled. This would create a nonequilibrium situation in which the coupled protons and the leak protons have different reversal potentials, and the observed zero-flux voltage lies between the two. In such a case of partial slippage; a situation known to exist, for example, in mutants of CLC transporters, the actual mechanism of transport would be identical for both di- and tripeptides, and so it would be invalid to tout this as an example of “one transporter, two mechanisms.”

A fairly simple and quick experiment could plug this logical hole (if the result comes out in the hoped-for way). If dipeptide transport is truly coupled to more protons than tripeptide transport (5 vs 3, say), then radiolabeled dipeptide should be accumulated to far higher steady levels than tripeptide for the same proton electrochemical gradient. If, on the other hand, leak protons are wasted while the mechanism's coupling is the same for both substrates, then di and tri-peptides should be similarly accumulated in a basic concentrative uptake assay. Even a qualitative result that could settle the issue.

2) This assay seems to work fine for two neutral tri-peptides, A-A-A and A-L-A. But the results are puzzlingly ambiguous for di-peptides that are known to be better substrates than tri-peptides. It seems that the di-peptides have a different stoichiometry than that of the tri-peptides; however, the authors could not determine the exact reversal potential for the di-peptides.

---

## [Author Response]

We performed the key experiment suggested by the reviewers to test our model of different stoichiometries for different substrates, namely a test of whether the peptide transporter, PepT_St_, could accumulate di-peptides to a higher concentration than tri-peptides for the same proton electrochemical gradient. The result of this experiment indeed confirms that PepT_St_ does indeed operate via two thermodynamically distinct mechanisms of peptide transport, and strengthens our conclusions. We have detailed the results in a new Figure 5.

*1) […] A fairly simple and quick experiment could plug this logical hole (if the result comes out in the hoped-for way). If dipeptide transport is truly coupled to more protons than tripeptide transport (5 vs 3, say), then radiolabeled dipeptide should be accumulated to far higher steady levels than tripeptide for the same proton electrochemical gradient. If, on the other hand, leak protons are wasted while the mechanism's coupling is the same for both substrates, then di and tri-peptides should be similarly accumulated in a basic concentrative uptake assay. Even a qualitative result that could settle the issue*.

We agree with the reviewers that this is an important experiment and as detailed above we have included this data in a new figure, Figure 5. The results clearly show that di-alanine peptide is concentrated to a significantly higher level than tri-alanine for the same proton electrochemical gradient. We are confident that this new result substantially strengthens our original hypothesis and the conclusions in the paper.